# HGWM: Hierarchical Graph-guided World Model for Zero-shot Object Navigation via Scene-Goal Graph Matching

## Abstract

Object Goal Navigation, which requires an agent to locate specific objects in unknown indoor environments, remains a fundamental challenge in embodied AI that demands sophisticated spatial-semantic understanding. Although recent Vision-Language Model (VLM) based approaches have shown promise through effective perception and reasoning capabilities, current methods lack systematic world model architectures that can predict environmental states and reduce exploration inefficiency. We introduce HGWM (Hierarchical Graph-guided World Model), a novel navigation framework that integrates dual-graph matching with a unified Spatial-Semantic World Model to enable robust object localization. HGWM constructs complementary graph representations: a goal subgraph encoding LLM-derived spatial knowledge about target objects and room hierarchies, and dynamically maintained scene graphs derived from our persistent Spatial-Semantic World Model. These graphs interact through a dual-matching mechanism that combines implicit VLM-guided semantic alignment with explicit structural correspondence. Our multi-stage exploration strategy adapts dynamically based on the degree of graph matching, transitioning from systematic exploration to focused search and finally target verification. Experiments on HM3D v0.1, v0.2, and MP3D benchmarks demonstrate HGWM's effectiveness, achieving state-of-the-art performance with 59.6% success rate and 31.5% SPL on HM3D v0.1, and 45.8% success rate and 17.3% SPL on MP3D, outperforming previous methods by up to +1.5% SR and +0.3% SPL. Our code will be made public soon.

## 1 Introduction

Humans exhibit a remarkable ability to navigate through unfamiliar environments by leveraging a sophisticated internal model of the world. This model is not merely a collection of visual memories but a rich, hierarchical understanding of spatial and semantic relationships. For instance, we infer that a kitchen is likely near a dining area or that a bedroom is often connected to a private bathroom. This capacity for hierarchical spatial-semantic reasoning allows for efficient exploration and goal-finding, even in spaces encountered for the first time.

Inspired by human spatial reasoning, recent advances in zero-shot object navigation have leveraged foundation models such as LLMs (Achiam et al., 2023; Touvron et al., 2023) and VLMs to develop more sophisticated navigation systems (Yin et al., 2024; 2025; Yokoyama et al., 2024; Kuang et al., 2024). These training-free approaches offer valuable adaptability across platforms and interpretable reasoning steps. As comprehensive surveys show (Ieong & Tang, 2025; Sun et al., 2024a), current methods fall into two paradigms: embedding-based approaches using frozen vision-language models for semantic matching (Majumdar et al., 2022; Gadre et al., 2023; Yokoyama et al., 2024; Khandelwal et al., 2022), and linguistic inference methods employing LLMs for high-level spatial reasoning (Zhou et al., 2023; Kuang et al., 2024; Yin et al., 2024; Li et al., 2022). Despite their strengths, both paradigms struggle to construct coherent world models that effectively predict environmental states. Most approaches rely on either task-specific workflows limiting their general applicability or simplistic exploration strategies that fail to capture hierarchical indoor structures and rich contextual relationships between objects, resulting in inefficient navigation and suboptimal performance in complex environments.

To address these limitations, we introduce HGWM (Hierarchical Graph-guided World Model), a novel framework for zero-shot object goal navigation that constructs and maintains a rich, graph-based world model. Unlike existing approaches that treat embedding-based similarity matching and linguistic spatial reasoning as separate paradigms (Ieong & Tang, 2025), our approach moves beyond representing scenes and objectives as simple textual descriptions or isolated embeddings, instead adopting a unified graph-based formalism that systematically integrates both the agent's internal world model and the specified navigation goal. This consistent representation minimizes the loss of structural information and enables powerful, explicit reasoning through graph-based algorithms, including similarity computation and structural alignment.

At the core of our system is the dynamic construction of a 3D scene graph, which is continuously updated as the agent explores the environment. At each decision point, we employ a dual-graph matching mechanism that performs an explicit structural comparison between the dynamically built scene graph and a pre-computed goal subgraph, which encapsulates hierarchical knowledge about the target object and its likely spatial context, derived from an LLM. Concurrently, a scene analysis VLM provides implicit semantic alignment by analyzing panoramic visual observations to complement the structural matching process.

Navigation decisions are guided by a multi-stage exploration policy that adapts its strategy based on the degree of correspondence between the scene and goal graphs. When the agent has minimal environmental information (zero matching), it prioritizes discovering new areas through systematic exploration. As contextual features of the goal environment are observed (partial matching), the strategy shifts to focused search, using anchor objects and coordinate projection to infer the target's likely location. Finally, upon achieving strong correspondence (perfect matching), the agent transitions to direct navigation and goal verification. This structured, multi-stage approach synthesizes explicit graph matching with implicit semantic guidance, enabling our agent to navigate with efficiency and robustness that significantly surpasses existing zero-shot methods.

We demonstrate HGWM's effectiveness on standard benchmarks including HM3D v0.1, HM3D v0.2, and MP3D datasets, where we achieve state-of-the-art performance compared to existing zero-shot object navigation methods. Specifically, our approach achieves a 59.6% success rate and a 31.5% SPL on HM3D v0.1, and a 45.8% success rate and a 17.3% SPL on MP3D, outperforming the previous best method WMNav (Nie et al., 2025) by +1.5% SR/+0.3% SPL and +0.4% SR/+0.1% SPL respectively. Our approach shows significant improvements in both success rate and path efficiency across diverse indoor environments, demonstrating the value of hierarchical graph-guided world models for object goal navigation tasks and advancing the state-of-the-art in zero-shot navigation capabilities.

## 2 RELATED WORK

**Zero-shot Object Goal Navigation.** Existing object navigation methods fall into two paradigms: supervised methods and zero-shot methods. Supervised approaches train visual encoders with RL/IL policies (Ramrakhya et al., 2022; Yadav et al., 2023), limiting their generalizability to unseen objects and environments. Zero-shot works address this via open-vocabulary scene understanding without requiring task-specific training. ZSON (Majumdar et al., 2022) pioneered multimodal goal embeddings for zero-shot navigation using CLIP embeddings (Khandelwal et al., 2022), while CoWs (Gadre et al., 2023) established important baselines for language-driven zero-shot object navigation. ESC (Zhou et al., 2023) introduced soft commonsense constraints leveraging grounded vision-language models (Li et al., 2022) to improve exploration efficiency. Recent approaches include L3MVN (Yu et al., 2023) and VoroNav (Wu et al., 2024) for large language model integration in visual target navigation. However, these zero-shot methods are fundamentally limited by their reliance on simple exploration strategies and lack of persistent environmental understanding. Most approaches treat navigation decisions independently without maintaining coherent spatial-semantic relationships across time steps.

**Foundation Model Guided Navigation.** Foundation models have revolutionized navigation by providing rich semantic understanding and spatial reasoning capabilities. Recent advances highlight the complementary roles of Vision-Language Models (VLMs) and Large Language Models (LLMs), driven by their distinct strengths in visual grounding and commonsense reasoning. Vision-language models like LLaVA (Liu et al., 2023) and grounded vision-language pre-training approaches (Li

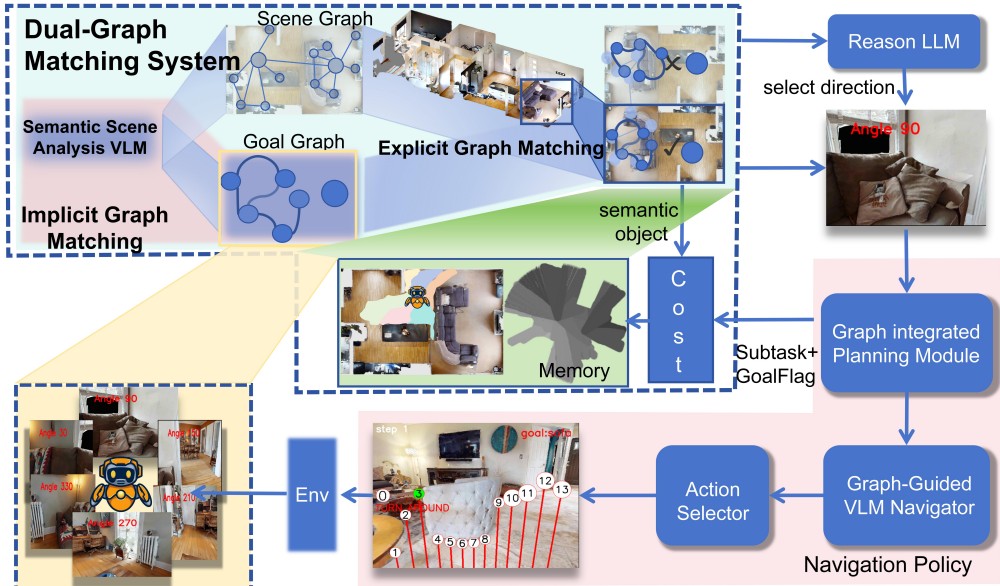

Figure 1: **The HGWM framework.** Upon receiving RGB-D observations and a target goal, the system first uses an LLM to construct a hierarchical Goal Subgraph capturing spatial-semantic knowledge about the target. Next, the Spatial-Semantic World Model processes incoming observations by precisely anchoring detected objects to 3D coordinates while preserving their semantic relationships. The Dynamic Scene Graph Maintenance component continuously updates and refines the observed scene graph based on new information. Finally, the Dual-Graph Matching System performs both implicit VLM-guided alignment and explicit structural comparison between the observed scene graph and goal subgraph to determine the optimal navigation direction. This direction is then translated into concrete navigation actions through an action selector. The entire process leverages both geometric spatial understanding and semantic knowledge to effectively navigate toward the target object.

et al., 2022) have demonstrated strong capabilities in embodied AI tasks (Khandelwal et al., 2022). World models have shown great promise by providing predictive capabilities and reducing exploration inefficiency. PathDreamer (Koh et al., 2021) introduced world models specifically for indoor navigation, while WMNav (Nie et al., 2025) demonstrates the effectiveness of integrating vision-language models with world models for object goal navigation. Graph representations have emerged as powerful tools for capturing spatial relationships and enabling structured reasoning. SG-Nav (Yin et al., 2024) introduced semantic graph representations for LLM-based navigation, while SayPlan (Rana et al., 2023) grounds LLMs using 3D scene graphs for scalable task planning. UniGoal (Yin et al., 2025) developed unified goal representations that combine visual and semantic information in graph-structured formats. Traditional world model approaches require extensive training and do not leverage semantic foundation model knowledge effectively. Graph-based methods typically operate on static graphs without dynamic updating capabilities during exploration. Most critically, current approaches lack systematic integration of zero-shot foundation model capabilities with structured spatial reasoning and persistent world modeling. Our HGWM framework addresses these limitations by combining zero-shot foundation model reasoning with systematic graph-based world modeling and dynamic scene understanding.

## 3 THE PROPOSED METHOD

In this section, we first define the object goal navigation problem and establish our mathematical framework. We then introduce our Hierarchical Graph-guided World Model (HGWM) architecture, which extends the WMNav foundation by integrating rich semantic representations with a persistent world model (see Figure 1 for a system overview). Following this, we detail our Spatial-Semantic World Model, the innovative dual-graph matching system, and finally, our LLM-guided spatial reasoning and exploration strategy.

## 3.1 PROBLEM FORMULATION

We formulate object goal navigation as a partially observable Markov decision process (POMDP) where an agent must locate an instance of a specified object category $c$ (e.g., bed, sofa) in an unknown environment. At each time step $t$, the agent receives an RGB-D observation $O_t = \{I_t \in \mathbb{R}^{H \times W \times 3}, D_t \in \mathbb{R}^{H \times W}\}$ and its pose $P_t = (x_t, y_t, \theta_t)$, then selects an action $a_t = (r_t, \theta_t)$ in polar coordinates, representing direction and travel distance. This continuous action space enables more flexible navigation compared to the discrete actions commonly used in prior work. Success is achieved when the agent stops within a threshold distance $d_{thres}$ (typically 1.0m) of any target category instance, with the objective of minimizing steps $T$ while maximizing the success rate across environments.

## 3.2 HIERARCHICAL GRAPH-GUIDED WORLD MODEL ARCHITECTURE

HGWM (Hierarchical Graph-guided World Model) extends the WMNav (Nie et al., 2025) foundation by integrating rich semantic understanding with spatial mapping through a novel hierarchical graph-based architecture. Unlike previous methods that rely solely on visual features or simplistic world models, HGWM constructs and maintains two complementary graph structures—a goal subgraph and a dynamic scene graph—which are grounded in a unified Spatial-Semantic World Model. This model serves as the spatial-semantic foundation for all graph operations, enabling persistent memory and sophisticated reasoning capabilities.

**Goal Subgraph Construction.** For a given goal category $c_{target}$, we leverage an LLM to construct a hierarchical goal subgraph $G_{goal}$ that encapsulates spatial-semantic knowledge about target environments:

$$G_{goal} = \text{LLM}_{\text{construct}}(c_{target}, \mathcal{P}_{spatial}). \tag{1}$$

The resulting $G_{goal}$ contains three complementary components: (1) **Room Hierarchy** ($H_{goal}$) classifying spaces into primary, secondary, and adjacent zones based on target object relevance; (2) **Spatial Anchors** ($V_{goal}$) identifying landmark objects that serve as navigation waypoints; and (3) **Room Connectivity** ($E_{goal}$) defining spatial transitions between room types. This structured representation transforms abstract spatial knowledge into a navigable hierarchy for systematic exploration (see Appendix A for detailed prompt design).

**Spatial-Semantic World Model.** A fundamental innovation in HGWM is our Spatial-Semantic World Model ($M_{world}$), which integrates spatial and semantic information into a unified, persistent representation. This component serves as the foundational memory system that anchors detected objects to precise 3D coordinates while preserving their semantic attributes and relational context.

For each panoramic observation, objects detected by the VLM are projected into the world coordinate system through depth-based mapping:

$$P(o_i) = \text{Project}(o_i, D_t, P_t), \tag{2}$$

where $o_i$ represents a detected object, $D_t$ provides depth information, and $P_t$ denotes the agent's current pose. The world model is then dynamically updated with both spatial and semantic information:

$$M_{world}[\text{key}(P(o_i))] \leftarrow \{o_i, \text{type}(o_i), \text{room}(o_i), \text{conf}(o_i)\}, \tag{3}$$

where $\text{key}(P(o_i))$ generates a unique spatial identifier for the object's 3D location. This integrated mapping system enables persistent spatial reasoning about objects and environments, even when they are no longer in the agent's immediate field of view, forming the foundation for dynamic scene graph construction and long-term spatial memory.

**Dynamic Scene Graph Maintenance** Our system maintains a semantically rich, hierarchical scene graph $G_{scene}$ that evolves dynamically during navigation. For each potential exploration direction $\alpha$, we construct a direction-specific local graph $G_{scene}^{\alpha}$ that captures the spatial-semantic structure of the environment in that direction.

Each direction-specific graph contains three types of information:

- **Object Nodes**: Detected objects with positional categorization based on their spatial distribution and semantic properties

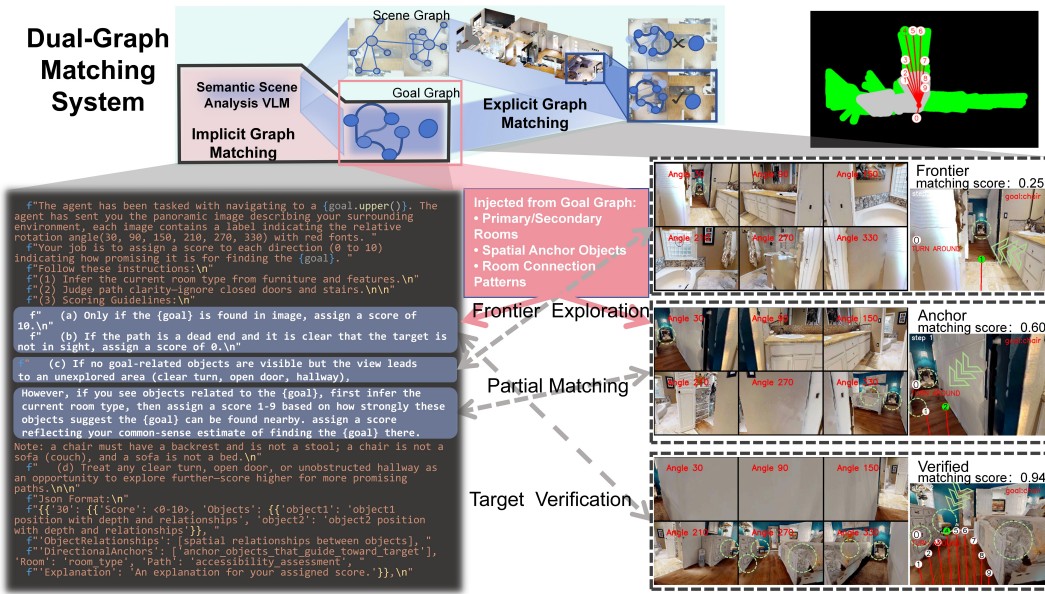

Figure 2: **Illustration of dual-graph matching system.** The figure demonstrates how our dual-graph matching mechanism operates through both implicit VLM-guided semantic alignment and explicit structural correspondence between the dynamically constructed scene graph and the goal subgraph during a navigation episode.

- **Room Classifications**: Space-type predictions with associated confidence scores derived from observed objects and spatial layout
- **Spatial Relationship Edges**: Connections between objects that encode positional compatibility and semantic relationships

The construction process is formalized as:

$$G_{scene}^{\alpha} = \text{GraphBuild}(M_{world}, \alpha), \tag{4}$$

where the GraphBuild function extracts relevant spatial-semantic information from the world model for the specified direction. This hierarchical representation enables precise spatial reasoning about each potential navigation direction, providing the structured foundation necessary for our sophisticated graph matching algorithms.

**Dual-Graph Matching.** The core innovation of HGWM lies in its sophisticated dual-graph matching system, which integrates two complementary alignment processes between the local scene graph $G_{scene}^{\alpha}$ and the target goal subgraph $G_{goal}$ (see Figure 2 for an illustration). This dual approach combines the efficiency of implicit semantic reasoning with the precision of explicit structural analysis.

**Stage 1: Implicit VLM-guided Matching.** The first stage leverages a Vision-Language Model to conduct goal-aware panoramic analysis, providing rapid semantic assessment without the computational overhead of structural graph comparison. The VLM prompt is strategically enriched with high-level knowledge extracted from $G_{goal}$, including target room types, anchor objects, and spatial context:

$$\text{Scores}_{\text{implicit}} = \text{VLM}(I_{\text{panoramic}}, \text{Prompt}(G_{goal})). \tag{5}$$

This stage efficiently processes visual observations while leveraging semantic context from the goal graph, providing initial directional preferences based on visual-semantic alignment.

**Stage 2: Explicit Structural Graph Matching.** Following the implicit assessment, we perform a comprehensive five-stage hierarchical structural analysis that provides precise correspondence between the local scene graph $G_{scene}^{\alpha}$ and goal subgraph $G_{goal}$.

*Sub-stage 2.1: LLM-guided Scene Graph Correction.* We first apply intelligent scene graph corrections to address perceptual inconsistencies and implausible spatial arrangements that may arise from noisy visual detection:

$$G_{corrected} = \text{LLM}_{\text{correct}}(G_{scene}^{\alpha}, C_{goal}, S_{spatial}), \tag{6}$$

where $C_{goal}$ provides goal-specific context and $S_{spatial}$ represents spatial constraints derived from the goal subgraph.

*Sub-stage 2.2: Multi-dimensional Embedding-based Similarity.* Rather than relying on direct string matching, we compute comprehensive similarity scores through multiple specialized embedding-based components that capture semantic relationships across different dimensions. For each matching component (room, object, spatial, semantic, depth), we leverage dense embeddings to enable nuanced semantic understanding.

For instance, the object matching component computes similarity using cosine similarity in embedding space:

$$\text{Sim}(v_i, v_j) = \frac{\text{Embed}(v_i) \cdot \text{Embed}(v_j)^T}{||\text{Embed}(v_i)|| \cdot ||\text{Embed}(v_j)||}, \tag{7}$$

where $v_i$ and $v_j$ represent nodes from the corrected scene graph and goal graph, respectively. This cosine similarity approach enables sophisticated matching that captures semantic relationships beyond exact lexical correspondence.

*Sub-stage 2.3: Position-aware Enhancement.* We apply a position-aware enhancement function that strategically rewards partial matches exhibiting strong spatial coherence, particularly crucial when only portions of the target environment are visible during exploration:

$$B(G_{corrected}, G_{goal}) = \sum_{(v_i, v_j) \in \mathcal{M}} w_{spatial}(v_i, v_j) \cdot \text{Coherence}(v_i, v_j), \tag{8}$$

where $\mathcal{M}$ represents the set of matched node pairs and $\text{Coherence}(v_i, v_j)$ measures spatial consistency.

*Sub-stage 2.4: Comprehensive Integration.* The final explicit matching score integrates all specialized components with direction-specific factors:

$$S_{explicit}(\alpha) = \sum_i w_i \cdot S_i(G_{corrected}, G_{goal}) \cdot D(\alpha) \cdot R(\alpha) + B(G_{corrected}, G_{goal}), \tag{9}$$

where $R(\alpha)$ represents the direction relevance function, $D(\alpha)$ encodes position-direction compatibility, and the summation aggregates scores across all matching dimensions.

**Scene Graph-to-Decision Integration.** The updated scene graph representations from the LLM reasoning integration are transformed into actionable navigation decisions through a systematic integration process. The comprehensive directional data $\{\text{data}_\alpha\}_{\forall \alpha}$ incorporates the updated scene graph structures:

$$\text{data}_\alpha = \{S_{\text{implicit}}(\alpha), S_{\text{explicit}}(\alpha), G_{scene}^{\alpha,(t+1)}, \text{Context}_\alpha\}, \tag{10}$$

where $G_{scene}^{\alpha,(t+1)} = \{V_o^{(t+1)}, E_o^{(t+1)}\}$ represents the updated scene graph for direction $\alpha$, and $\text{Context}_\alpha$ includes spatial relationship hierarchies and confidence distributions derived from the graph reasoning process.

**Integrated Decision Framework.** The dual matching processes are synthesized through sophisticated LLM-guided reasoning that holistically integrates the updated scene graph information with implicit semantic scores and explicit structural correspondence. This comprehensive integration process is formalized as:

$$\alpha^*, \text{confidence} = \text{LLM}_{\text{reason}}(\{\text{data}_\alpha\}_{\forall \alpha}, G_{\text{goal}}, \text{ExplorationPhase}). \tag{11}$$

The LLM reasoning function receives enriched directional data that includes: (1) implicit VLM semantic scores, (2) explicit structural matching results, (3) updated scene graph representations with refined node and edge attributes, (4) goal graph context, and (5) current exploration phase indicators.

The LLM reasoning component $S_{\text{llm}}(\alpha)$ synthesizes all this information to produce direction-specific scores that incorporate both immediate visual-semantic alignment and long-term spatial-structural coherence:

$$S_{\text{llm}}(\alpha) = \text{LLM}_{\text{integrate}}(G_{scene}^{\alpha,(t+1)}, S_{\text{implicit}}(\alpha), S_{\text{explicit}}(\alpha), G_{\text{goal}}). \tag{12}$$

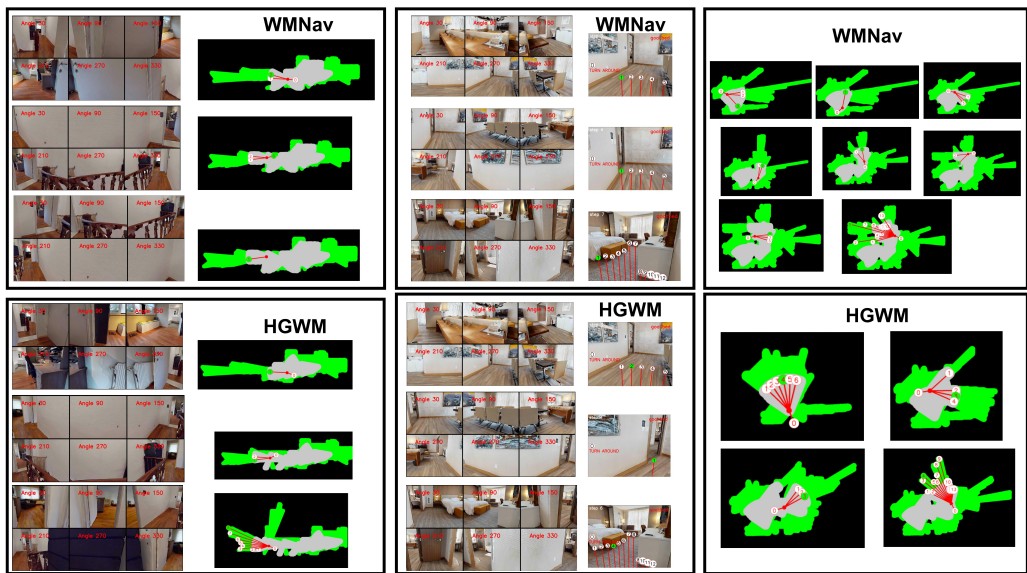

Figure 3: **Decision process comparison between WMNav and HGWM.** HGWM leverages graph-based memory to avoid repetitive exploration in the same space and encourages the agent to move toward regions with greater object visibility, resulting in more efficient and diverse exploration trajectories compared to WMNav.

The final navigation decision combines spatial navigation metrics with this comprehensive LLM reasoning through a streamlined framework:

$$\text{Score}_{\text{final}}(\alpha) = w_{\text{spatial}} \cdot S_{\text{base}}(\alpha) + w_{\text{llm}} \cdot S_{\text{llm}}(\alpha). \tag{13}$$

This integrated approach ensures seamless information flow from scene graph updates through comprehensive reasoning to final navigation decisions, leveraging both the semantic intuition of VLMs and the structural rigor of graph matching for robust navigation across varying environmental conditions and partial observability scenarios.

### 3.3 LLM-Guided Multi-Stage Reasoning

Based on the dual-graph matching results and overlap scores between $G_{goal}$ and $G_{scene}$, our system implements an adaptive three-stage exploration strategy: **Frontier Exploration** (zero matching - prioritizes systematic exploration), **Focused Search** (partial matching - balances exploration-exploitation using anchor objects), and **Target Verification** (perfect matching - focuses on precise goal localization). The LLM reasoning component integrates scene graph structural information with directional assessment, dynamically updating node and edge representations while synthesizing implicit VLM scores with explicit structural correspondence to generate optimal navigation decisions. The detailed formulation and implementation of this multi-stage reasoning framework are provided in Appendix C.

### 3.4 Navigation Policy

HGWM adopts and extends WMNav's hierarchical navigation architecture while integrating rich graph-based information at each decision level. Our navigation policy consists of graph-guided subtask decomposition and a two-stage action selection mechanism, both enhanced with structural graph knowledge. For subtask decomposition, we leverage the hierarchical goal subgraph to identify semantically meaningful intermediate targets that efficiently lead to the final goal. Rather than reasoning solely from visual observations, our PlanVLM receives augmented prompts containing graph-derived spatial-semantic context: $\text{SubTask}_t, \text{GoalFlag}_t = \text{PlanVLM}(I_t(\alpha), \text{SubTask}_{t-1}, G_{scene}^{\alpha,(t)}, G_{goal})$. This graph-enriched planning enables the agent to generate more informed subtasks based on structural scene understanding (e.g., "go through the living room doorway" rather than just "go forward") by recognizing room transitions and spatial landmarks from the scene graph.

Table 1: Comparison with state-of-the-art object navigation methods on HM3D v0.1 and MP3D benchmarks. US denotes Unsupervised, ZS denotes Zero-Shot capabilities, and II denotes Instruction Interpolator.

| Method | US | ZS | Vision | Language | HM3D v0.1 | | MP3D | |
|---|---|---|---|---|---|---|---|---|
| | | | | | SR↑ | SPL↑ | SR↑ | SPL↑ |
| Habitat-Web (Ramrakhya et al., 2022) | ✗ | ✗ | - | - | 41.5 | 16.0 | 31.6 | 8.5 |
| OVRL (Yadav et al., 2023) | ✗ | ✗ | - | - | - | - | 28.6 | 7.4 |
| OVRL-V2 (Yadav et al., 2023) | ✗ | ✗ | - | - | 64.7 | 28.1 | - | - |
| ZSON (Majumdar et al., 2022) | ✗ | ✓ | CLIP | - | 25.5 | 12.6 | 15.3 | 4.8 |
| PSL (Sun et al., 2024b) | ✗ | ✓ | CLIP | - | 42.4 | 19.2 | 18.9 | 6.4 |
| PixNav (Cai et al., 2024) | ✗ | ✓ | LLaMA-Adapter | GPT-4 | 37.9 | 20.5 | - | - |
| SGM (Zhang et al., 2021) | ✗ | ✓ | - | - | 60.2 | 30.8 | 37.7 | 14.7 |
| VLFM (Yokoyama et al., 2024) | ✗ | ✓ | BLIP-2 | - | 52.5 | 30.4 | 36.4 | 17.5 |
| CoW (Gadre et al., 2023) | ✓ | ✓ | CLIP | - | - | - | 9.2 | 4.9 |
| ESC (Zhou et al., 2023) | ✓ | ✓ | - | GPT-3.5 | 39.2 | 22.3 | 28.7 | 14.2 |
| L3MVN (Yu et al., 2023) | ✓ | ✓ | - | GPT-2 | 50.4 | 23.1 | - | - |
| VoroNav (Wu et al., 2024) | ✓ | ✓ | BLIP | GPT-3.5 | 42.0 | 26.0 | - | - |
| OpenFMNav (Kuang et al., 2024) | ✓ | ✓ | - | - | 54.9 | 24.4 | - | - |
| TopV-Nav (Zhong et al., 2024) | ✓ | ✓ | - | - | 45.9 | 28.0 | 31.9 | 16.1 |
| SG-Nav (Yin et al., 2024) | ✓ | ✓ | LLaVA-1.6-7B | GPT-4 | 54.0 | 24.9 | 40.2 | 16.0 |
| UniGoal (Yin et al., 2025) | ✓ | ✓ | LLaVA-1.6-7B | LLaMA-2-7B | 54.5 | 25.1 | 41.0 | 16.4 |
| WMNav (Nie et al., 2025) | ✓ | ✓ | Gemini-1.5-Pro | - | 58.1 | 31.2 | 45.4 | 17.2 |
| **HGWM (Ours)** | ✓ | ✓ | **Gemini-1.5-Pro** | **Qwen2.5-7B-Instruct** | **59.6** | **31.5** | **45.8** | **17.3** |

The action selection process similarly benefits from graph integration through our dual-stage approach. In the exploration stage, candidate actions $A_t^{cand} = \{(r_{t,j}, \theta_{t,j})\}_{j=1}^{K'}$ are filtered not only by the exploration state map but also through graph-based relevance scoring, prioritizing directions with higher graph matching scores from our dual-graph matching system. The ActionVLM receives scene graph annotations alongside visual observations: $a_t = $ ReasonVLM(SubTask$_t$, Goal$_t$, $I_t^{ann}$, $G_{scene}^{\alpha,(t)}$). When transitioning to the goal-approaching stage, our system leverages position-aware graph matching to locate the target more precisely, utilizing object-relation context from the scene graph to improve goal localization accuracy. This integrated approach ensures that navigation decisions benefit from both immediate visual cues and structured spatial-semantic understanding, resulting in more coherent and efficient navigation trajectories.

# 4 EXPERIMENTS

We conduct comprehensive experiments on three standard benchmarks to evaluate our Hierarchical Graph-guided World Model (HGWM) approach for zero-shot object navigation. Our experimental design focuses on demonstrating the effectiveness of graph-guided reasoning in diverse indoor environments while maintaining unsupervised and zero-shot capabilities.To further illustrate the advantages of our graph memory mechanism, Figure 3 compares the decision-making processes of WMNav and HGWM across representative episodes. Unlike WMNav, which often revisits the same areas and exhibits looped exploration, HGWM utilizes its structured memory to promote exploration of new spaces and prioritizes directions that maximize object visibility. This leads to more efficient coverage and reduces redundant navigation, demonstrating the effectiveness of graph-guided reasoning in complex environments.

## 4.1 DATASETS AND EVALUATION METRICS

**Datasets.** We evaluate our method on three widely-used benchmarks: (1) **HM3D v0.1** (Ramakrishnan et al., 2021) is used in the Habitat 2022 ObjectNav challenge, providing 2000 validation episodes across 20 validation environments with 6 goal object categories. (2) **HM3D v0.2** (Ramakrishnan et al., 2021) represents the improved version of HM3D with enhanced geometry and semantic labels, including typo fixes and painting error corrections, containing 1000 validation episodes with higher quality annotations. (3) **MP3D** (Chang et al., 2017) contains 11 high-fidelity photorealistic scenes with 2195 validation episodes covering 21 categories of object goals, providing a challenging benchmark for navigation methods.

**Evaluation Metrics.** We adopt two standard metrics: *Success Rate* (SR) represents the percentage of episodes completed successfully, and *Success Rate Weighted by Inverse Path Length* (SPL) quantifies navigation efficiency by calculating the inverse ratio of actual path length to optimal path length, weighted by success rate. These metrics provide a comprehensive evaluation of both navigation effectiveness and efficiency.

Table 2: Component contribution analysis on HM3D v0.2 (Ramakrishnan et al., 2021).

| Method | Memory Structure | Graph Matching | SR(%)↑ | SPL(%)↑ |
|---|---|---|---|---|
| Basic VLM Nav | ✗ No Memory | Simple Scoring | 61.8 | 29.3 |
| VLM + Memory | ✓ Voxel Map | Simple Scoring | 71.9 | 32.8 |
| **HGWM (Ours)** | ✓ Hierarchical Graph | Dual-graph | **72.4** | **32.9** |

To further illustrate the advantages of our graph memory mechanism, Figure 3 compares the decision-making processes of WMNav and HGWM across representative episodes. Unlike WM-Nav, which often revisits the same areas and exhibits looped exploration, HGWM utilizes its structured memory to promote exploration of new spaces and prioritizes directions that maximize object visibility. This leads to more efficient coverage and reduces redundant navigation, demonstrating the effectiveness of graph-guided reasoning in complex environments.

## 4.2 COMPARISON WITH STATE-OF-THE-ART METHODS

HGWM achieves state-of-the-art performance across both benchmarks (Table 1). On HM3D v0.1, we achieve 59.6% SR and 31.5% SPL, outperforming the previous best method WMNav (Nie et al., 2025) by +1.5% SR and +0.3% SPL. On MP3D, we achieve 45.8% SR and 17.3% SPL, with +0.4% SR and +0.1% SPL improvements over WMNav.

Compared to learning-based methods like OVRL-V2 (Yadav et al., 2023), our approach provides zero-shot generalization without requiring task-specific training. Among zero-shot methods, we significantly outperform ZSON (Majumdar et al., 2022), VLFM (Yokoyama et al., 2024), and graph-based approaches like SG-Nav (Yin et al., 2024) and UniGoal (Yin et al., 2025), demonstrating the effectiveness of our hierarchical graph-guided world model framework for structured spatial reasoning in complex navigation scenarios.

## 4.3 ABLATION STUDIES

We analyze the contribution of each HGWM component on HM3D v0.2 (Ramakrishnan et al., 2021) (Table 2). Basic VLM navigation without memory achieves 61.8% SR and 29.3% SPL. Adding voxel-based memory significantly improves performance by +10.1% SR and +3.5% SPL, demonstrating that spatial persistence enhances navigation effectiveness. Our full hierarchical graph memory with dual-graph matching (HGWM) achieves 72.4% SR and 32.9% SPL, with +0.5% SR and +0.1% SPL gains over voxel-based memory, confirming that structured spatial-semantic relationships offer advantages in complex environments requiring sophisticated reasoning.

## 5 CONCLUSION

We introduce HGWM (Hierarchical Graph-guided World Model), which establishes a novel paradigm for zero-shot object goal navigation by integrating dual-graph matching mechanisms with a unified Spatial-Semantic World Model, achieving state-of-the-art performance across multiple challenging benchmarks. Our method addresses fundamental limitations in existing approaches through systematic integration of complementary graph representations: LLM-derived goal sub-graphs capturing hierarchical spatial knowledge and dynamically maintained scene graphs from persistent spatial-semantic memory. The core innovation lies in our multi-dimensional explicit structural graph matching that combines implicit VLM-guided semantic alignment with explicit structural correspondence, utilizing embedding-based similarity computation and position-aware enhancement functions to handle partial observability scenarios. Experimental validation across HM3D v0.2 and MP3D datasets confirms HGWM's superiority over existing zero-shot methods, with comprehensive ablation studies revealing that each component—from hierarchical graph memory structures to dual-graph matching mechanisms—contributes meaningfully to overall performance. By establishing graph-based world models as a fundamental architectural component for embodied AI, HGWM opens new pathways for developing sophisticated navigation systems that reason about complex spatial-semantic relationships in unknown environments, with future directions including multi-target scenarios, temporal dynamics integration and broader embodied AI applications requiring structured environmental understanding and goal-directed behavior.

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

## A  GOAL SUBGRAPH CONSTRUCTION DETAILS

### A.1  LLM PROMPT TEMPLATE

We present the complete prompt template used for goal subgraph construction. The template is dynamically instantiated for each target object category, with {goal} replaced by the specific target (e.g., "bed", "sofa").

```
You are an expert in residential navigation and spatial
understanding.  Create a comprehensive navigation strategy
for efficiently locating a {GOAL} in a typical home
environment.

NAVIGATION STRATEGY ANALYSIS FOR {GOAL}:

1.  TARGET ROOM ANALYSIS:
(1) Primary room:  The most likely room type where {goal} is
typically found
(2) Secondary rooms:  1-2 alternative room types where
{goal} might also be located
(3) Adjacent zones:  Connected areas that provide visual
access or pathways to target rooms

2.  PATHWAY and CONNECTIVITY MAPPING:
(1) Hallway connections:  Common routes to reach target
rooms from main areas
(2) Visual access points:  Doorways/openings that help
identify target rooms
(3) Navigation landmarks:  Key reference points for
wayfinding

3.  TARGET-ORIENTED SPATIAL ANCHORS and NAVIGATION CUES:
(1) Directional anchors:  3-4 objects that DIRECTLY indicate
where {goal} is likely located (not just co-occurring
objects)
(2) Visual pathways:  Objects that form visual lines leading
toward typical {goal} placement areas
(3) Distance indicators:  Objects that help estimate
proximity to {goal} based on typical room layouts
(4) Positioning cues:  Furniture/fixtures that indicate the
specific area within a room where {goal} is usually found

4.  SYSTEMATIC EXPLORATION APPROACH:
(1) Exploration priorities:  Which areas to check first when
searching
(2) Recovery strategies:  What to do when initial search
paths fail
```

This structured prompt enables the LLM to generate hierarchical spatial knowledge encompassing room classifications, connectivity patterns, and object-based navigation cues, which are then parsed into the formal goal subgraph representation $G_{goal} = (H_{goal}, V_{goal}, E_{goal})$.

## B  IMPLEMENTATION DETAILS

**Agent Configuration.** We configure the agent with a maximum of 40 navigation steps per episode. The agent adopts a cylindrical body with radius 0.18m and height 0.88m, equipped with an ego-centric RGB-D camera at 640×480 resolution and 79° horizontal field of view. The camera is tilted down with a 14° pitch to improve navigable area identification. The success threshold $d_{thres}$ is set to 1.0m, meaning episodes are successful when the agent stops within 1.0m of any target object instance. We primarily use Gemini VLM for our experiments due to its cost-effectiveness and high performance in spatial reasoning tasks.

## C  DETAILED MULTI-STAGE REASONING FRAMEWORK

### C.1  MULTI-STAGE EXPLORATION STRATEGY

Our navigation system employs three distinct exploration phases, each optimized for different levels of goal-environment correspondence:

**Frontier Exploration Phase.** Activated when minimal structural matching is detected between $G_{goal}$ and $G_{scene}$ (overlap $< 0.3$), indicating limited knowledge about the current environment. During this phase, the system prioritizes systematic exploration strategies with weights heavily favoring spatial navigation and coverage over semantic alignment. The agent focuses on discovering new areas and building comprehensive environmental understanding.

**Focused Search Phase.** Engaged when moderate matching signals are detected ($0.3 \leq$ overlap $< 0.7$), suggesting the presence of goal-relevant environmental features. This phase employs balanced weights to achieve optimal exploration-exploitation trade-offs while actively seeking directions that demonstrate promising semantic similarity with the target environment. The system begins to leverage discovered anchor objects and spatial relationships for directed search.

**Target Verification Phase:** Triggered when strong matching correspondence is identified (overlap $\geq 0.7$), indicating a high likelihood of goal proximity. This phase applies comprehensive LLM-based scene graph correction to refine perceptual uncertainties and prioritizes precise localization over broad exploration. The agent concentrates on detailed verification and accurate goal identification.

### C.2  LLM-SCENE GRAPH REASONING INTEGRATION

The LLM reasoning component performs sophisticated analysis by deeply integrating scene graph structural information with directional assessment capabilities. This integration enables dynamic updating of both node and edge representations:

$$V_o^{(t+1)} = \text{LLM}(A \cdot V_o^{(t)}, M \cdot E_o^{(t)}, \text{VLM}(I^{(t)})) \tag{14}$$

$$E_o^{(t+1)} = \text{LLM}(M^T \cdot V_o^{(t)}, A' \cdot E_o^{(t)}, \text{VLM}(I^{(t)})) \tag{15}$$

where $V_o$ and $E_o$ represent nodes and edges within distance $d$ of the target object $o$, while $A$ and $M$ denote adjacency matrices encoding spatial and semantic relationships.

The LLM receives comprehensive scene graph summaries including spatial relationship hierarchies, object classification hierarchies, and confidence score distributions. This rich contextual information enables sophisticated decision-making that can strategically prioritize directions with lower immediate matching scores but higher long-term exploration potential during frontier phases, or heavily weight precise object matches when targets are detected during verification phases.

## D  LLM USE DECLARATION

Large Language Models (ChatGPT) were used exclusively to improve the clarity and fluency of English writing. They were not involved in research ideation, experimental design, data analysis, or interpretation. The authors take full responsibility for all content.

