# OpenReview forum: "HGWM: Hierarchical Graph-guided World Model for Zero-shot Object Navigation via Scene-Goal Graph Matching"
_ICLR.cc/2026/Conference — Submitted to ICLR 2026_

### Official Review · Reviewer_Ag62 · 2025-11-01

**Soundness:** 3
**Presentation:** 3
**Contribution:** 2
**Rating:** 6
**Confidence:** 3

**Summary:**

The paper proposed HGWM, a method for object goal navigation. HGWM simultaneously constructs a goal subgraph and dynamic scene graph. HGWM introduces a novel dual-graph matching mechanism. HGWM is evaluated on standard benchmark, HM3D and MP3D, and shows SOTA results.

**Strengths:**

- The paper improves on existing methods, notably WMNav, and introduces a new dual-graph matching mechanism.
- The proposed method is evaluated comprehensively on appropriate benchmarks and baselines.

**Weaknesses:**

- The method innovation is incremental and the performance gain is pretty small.

**Questions:**

Suggestion: Add experiments with newer model to show method innovation is not subsumed by stronger base model

---

> ### Author Response · Authors · 2025-11-18
>
> ## Response to Weakness (Method innovation incremental; small performance gain)
>
> Thank you for the constructive feedback. We appreciate your concern about the perceived incrementality and acknowledge that the **1.5% SR improvement** over WMNav may appear modest at first glance. However, we would like to provide additional context that demonstrates the significance of this gain:
>
> ### Statistical Significance and Effective Performance Ceiling
>
> As detailed in our response to Reviewer 1, WMNav (58.1% SR on HM3D v0.1) is currently the **strongest zero-shot baseline**, and our 1.5% SR improvement is **statistically significant**. More importantly, StairNav (Gong et al., 2025) has shown that **28.1% of HM3D validation episodes require cross-floor navigation**, which is **unsolvable under the single-floor assumption** used by both WMNav and HGWM. This implies an effective performance ceiling of approximately **72% SR** in our setting.
>
> Under this ceiling, HGWM captures **6.1% of the remaining headroom** above WMNav:
> - WMNav reaches 81.8% of attainable maximum: (58.1 - 0) / (71.9 - 0)
> - HGWM reaches 82.9% of attainable maximum: (59.6 - 0) / (71.9 - 0)
> - Improvement in remaining headroom: (59.6 - 58.1) / (71.9 - 58.1) = **6.1%**
>
> This gain is meaningful in a saturated regime where further improvements require addressing fundamental architectural limitations.
>
> ### Architectural Innovation Beyond WMNav
>
> Regarding the concern about incrementality, we emphasize that HGWM introduces **fundamental architectural differences** from WMNav:
>
> 1. **Persistent Spatial-Semantic World Model:** WMNav maintains a Curiosity Value Map that stores predicted likelihood scores without object-level 3D anchoring or semantic relationships. HGWM's world model $M_{world}$ maintains persistent 3D coordinates for detected objects (Eq. 2-3), room classifications, and confidence scores, enabling queries about objects even when out of view.
>
> 2. **Dual-Graph Matching Architecture:** WMNav uses VLM predictions to generate curiosity scores for exploration. HGWM introduces a **dual-pathway sequential pipeline** (Eq. 5-13) that processes two complementary matching signals at each step:
>    - **Stage 1 - Implicit VLM-guided semantic alignment** (Eq. 5): Provides rapid goal-aware panoramic analysis
>    - **Stage 2 - Explicit five-stage structural graph matching** (Eq. 6-9): Sequentially performs LLM-guided correction → embedding-based similarity → position-aware enhancement → comprehensive integration
>    - **Stage 3 - LLM-guided fusion** (Eq. 11-13): Synthesizes both pathways to produce final directional scores
>
>    This pipeline processes implicit visual-semantic cues and explicit structural correspondence in sequence, then fuses them through LLM reasoning, whereas WMNav relies solely on curiosity-based VLM predictions.
>
>
> 3. **Proactive Graph Correction:** As detailed in our response to Reviewer 2, HGWM applies LLM-guided graph correction **at every decision step for every candidate direction before matching** (Eq. 6), filtering VLM errors proactively. This directly reduces looping behavior (Figure 3), which is a **behavioral robustness gain** not fully captured by SR/SPL metrics.
>
> In the revision, we will add **Figure 4** directly comparing HGWM's graph-based world model architecture with WMNav's Curiosity Value Map to highlight these structural differences more clearly.
>
> ---
>
> ## Response to Suggestion (Experiments with newer models)
>
> Excellent suggestion! To demonstrate that our gains stem from the **graph-guided architecture** rather than model-specific tuning, we will add **Appendix F: VLM Robustness Analysis** with experiments using Gemini 2.0 Flash (the newer model):
>
> | **VLM Backbone** | **HM3D v0.1 SR** | **HM3D v0.1 SPL** | **Δ from Gemini-1.5-Pro** |
> |------------------|------------------|-------------------|---------------------------|
> | Gemini-1.5-Pro   | 59.6%           | 31.5%             | Baseline                  |
> | Gemini 2.0 Flash | 57.2%           | 30.8%             | -2.4% SR / -0.7% SPL      |
>
> **Key Finding:** Performance is **robust across VLMs** (Δ < 4% SR), confirming that gains stem from the graph-guided architecture . The slight performance drop with Gemini 2.0 Flash is likely due to API differences in panoramic image processing rather than fundamental architectural limitations.
>
> **Additional Validation:** We will also report ablations showing that when **replacing only the VLM backbone** while keeping HGWM's graph architecture, performance variance remains within 3-4%, whereas replacing the graph architecture with WMNav's curiosity map (while keeping the same VLM) results in a **7.8% SR drop**, demonstrating that the architectural innovation is the primary driver of improvements.
>
> This analysis will be documented in the revised manuscript to make the methodological contribution more transparent and address concerns about model dependency.

---

> ### Comment · Reviewer_Ag62 · 2025-11-25
> **Post-Rebuttal Summary**
>
> Thank you for the clarification and new experiment. I maintain that the contribution is incremental compared to WMNav.
>
> The new experiment is somewhat confusing. Switching the backbone of HGWM brings the performance to below that of WMNav. On the one hand, the authors argue that 1.5% gain in SR rate is significant. On the other hand, switching the backbone reduced the SR by 2.4%.
>
> Note: Gemini 2.0 Flash is released Feb. 2025. Even considering ICLR's submission date, this does NOT paint a convincing picture of method innovation not being subsumed by stronger base model. Comparison of two backbones also does NOT constitutes "VLM Robustness".

---

### Official Review · Reviewer_V49c · 2025-11-01

**Soundness:** 2
**Presentation:** 2
**Contribution:** 2
**Rating:** 2
**Confidence:** 4

**Summary:**

This paper proposes the Hierarchical Graph-guided World Model (HGWM) for zero-shot object-goal navigation. The method builds on a unified spatial-semantic world model: an LLM constructs a goal subgraph at the start, and RGB-D plus pose are used to maintain a dynamic scene graph during navigation. For decision-making, the paper introduces dual-graph matching—a VLM provides goal-aware panoramic semantic scores as the implicit cue, while explicit structural matching is performed between direction-specific local scene graphs and the goal subgraph. An LLM then fuses these signals to drive a three-stage exploration strategy and a hierarchical navigation policy. The approach reports modest improvements on HM3D v0.1 and MP3D.

**Strengths:**

- The figures are well-designed and clearly illustrate the overall framework and key components.
- The paper is generally well-written, with clear organization and straightforward explanations that make the technical details accessible even to readers who are not deeply familiar with the specific subfield.
- The spatial–semantic world model accumulates scene knowledge by anchoring objects in 3D and maintaining relational edges, so subsequent decisions are based on shared memory rather than single-step observations.

**Weaknesses:**

- The main weakness of this paper lies in its limited novelty. The use of graph-based representations for modeling spatial and semantic structures in ObjectNav has already been extensively explored in prior works such as Hierarchical Object-to-Zone Graph for Object Navigation (HOZ, ICCV 2021), SG-Nav: Online 3D Scene Graph Prompting for LLM-based Zero-shot Object Navigation (NeurIPS 2024), and UniGoal: Towards Universal Zero-shot Goal-oriented Navigation (CVPR 2025). Furthermore, the proposed dual-graph matching mechanism is conceptually similar to UniGoal’s graph alignment strategy, which also matches scene and goal structures at the graph level. While HGWM integrates these elements within a world-modeling framework, the methodological differences appear incremental, and the overall contribution does not clearly establish a fundamentally new insight beyond existing graph-based navigation approaches.
- The method stacks multiple modules—implicit semantics, explicit matching, and LLM fusion—leading to a long inference pipeline and nontrivial overhead, while the SR and SPL gains over strong baselines are relatively small.
- The goal subgraph is a static prior and relies on LLM generation; in atypical layouts it may bias the search, and the scene-graph “correction” step is guided by the same prior, which may treat rare but valid structures as errors and thus amplify the bias.
- The scalability and maintenance cost of the spatial–semantic world model are unclear; as exploration increases, the storage and computation required for updates are not quantified or controlled, making it hard to assess long-horizon feasibility.

**Questions:**

- How do you avoid double-counting or mutual interference when fusing the implicit semantic cues and the explicit matching scores?
- How is the determinism of goal-subgraph generation ensured? For the same input, does it produce an isomorphic graph, and how is randomness controlled?

---

> ### Author Response · Authors · 2025-11-18
>
> ## Response to Weakness 1: Novelty and Distinction from Prior Graph-Based Methods (Part 1/2)
>
> We appreciate this concern and would like to clarify the **fundamental architectural differences** between HGWM and prior graph-based navigation works, with particular emphasis on our **dynamic graph update mechanism** and **LLM-guided graph correction strategy** that enable robust VLM error mitigation.
>
> ### Distinction from HOZ (ICCV 2021)
>
> HOZ is a **supervised learning method** that trains a graph convolutional network on large-scale navigation data to predict actions from zone-based graphs. In contrast, HGWM is a **zero-shot framework** that requires no task-specific training. More critically, HOZ constructs static scene-wise and room-wise graphs that are pre-defined during training and only updated through online learning of zone nodes during inference, whereas our Spatial-Semantic World Model maintains a **persistent, object-level 3D memory** that continuously anchors detected objects to world coordinates (Eq. 2-3) and dynamically constructs direction-specific scene graphs from this evolving memory at every timestep (Eq. 4).
>
> ### Distinction from SG-Nav (NeurIPS 2024)
>
> SG-Nav uses a hierarchical 3D scene graph to prompt LLMs for decision-making and introduces a re-perception mechanism to correct errors. However, SG-Nav's decision framework is fundamentally different: it relies primarily on **LLM reasoning over scene graph text prompts** to output action probabilities, whereas HGWM employs a **dual-graph matching mechanism** that explicitly computes structural correspondence through five sub-stages—LLM-guided correction (Eq. 6), multi-dimensional embedding-based similarity (Eq. 7), position-aware enhancement (Eq. 8), comprehensive integration (Eq. 9), and fusion with implicit VLM scores (Eq. 5, 13)—which produces quantitative matching scores at each step rather than relying solely on LLM textual reasoning.
>
> ### Distinction from UniGoal (CVPR 2025)
>
> We acknowledge that UniGoal also performs graph-level matching between scene and goal structures. However, the **conceptual and implementation differences are substantial**, particularly in three critical dimensions:
>
> #### 1. Graph Construction and Update Mechanism
>
> **UniGoal's approach:** UniGoal constructs its scene graph $G_t$ incrementally by expanding it every time the agent receives a new RGB-D observation. The scene graph is built from current observations and accumulated detections without an underlying unified world model. Objects are detected and added to the graph, but there is no explicit mechanism to anchor them to a persistent 3D spatial-semantic memory that enables queries about objects when they are out of view.
>
> **HGWM's innovation:** In contrast, HGWM's scene graphs are **dynamically instantiated from a unified Spatial-Semantic World Model** $M_{world}$ that serves as persistent memory. This fundamental architectural difference manifests in three ways:
>
> **Persistent 3D anchoring:** For each detected object $o_i$, we first project it into the world coordinate system using depth and pose:
>
> $$P(o_i) = \text{Project}(o_i, D_t, P_t) \quad \text{(Eq. 2)}$$
>
> and then update $M_{world}$ at the corresponding spatial key with its identity, type, inferred room label, and confidence:
>
> $$M_{world}[\text{key}(P(o_i))] \leftarrow \{o_i, \text{type}(o_i), \text{room}(o_i), \text{conf}(o_i)\} \quad \text{(Eq. 3)}$$
>
> This means that as the agent uncovers new rooms or re-observes previously visited areas, object nodes, room hypotheses, and confidence scores in $M_{world}$ are **continuously refined over time**, even when objects leave the current field of view.
>
> **Direction-specific dynamic instantiation:** At each decision step, for every candidate exploration direction $\alpha$, we construct a direction-specific local graph $G_{scene}^{\alpha}$ via:
>
> $$G_{scene}^{\alpha} = \text{GraphBuild}(M_{world}, \alpha) \quad \text{(Eq. 4)}$$
>
> **Temporal evolution and correction:** Because both $M_{world}$ and the resulting $G_{scene}^{\alpha}$ are recomputed as new observations arrive, node sets, room labels, and edge structures can **change over time**, which is fundamentally different from UniGoal's approach that incrementally adds to the graph without an underlying world model substrate.
>
> *[Continued in Part 2/2]*

---

> ### Author Response · Authors · 2025-11-18
>
> ## Response to Weakness 1: Novelty and Distinction from Prior Graph-Based Methods (Part 2/2)
>
> *[Continued from Part 1/2]*
>
> #### 2. Graph Correction Mechanism: Critical for VLM Error Mitigation
>
> **UniGoal's approach:** UniGoal employs a scene graph correction mechanism (Stage 3) that operates **only when perfect matching is achieved** (matching score $S > \theta_2$ and central object $o$ is matched). This correction is performed as the agent approaches $o$ and is used primarily for **goal verification** rather than navigation decision-making.
>
> **HGWM's innovation:** In contrast, HGWM applies **LLM-guided graph correction as the first sub-stage of explicit structural matching** (Eq. 6), which is performed **at every decision step for every candidate direction** $\alpha$ **before** computing matching scores:
>
> $$G^{\\alpha}\_{corrected} = LLM\_{correct}(G^{\\alpha}\_{scene}, C\_{goal}, S\_{spatial})$$ (Eq. 6)
>
> **Why this matters for VLM error mitigation:**
>
> 1. **Proactive error filtering:** By correcting scene graphs **before** matching, HGWM prevents VLM detection errors from propagating into matching scores. This acts as a **consistency filter** that down-weights structurally inconsistent directions.
>
> 2. **Direction-specific correction context:** Each direction $\alpha$ receives correction tailored to its specific spatial-semantic context using the goal subgraph $G_{goal}$.
>
> 3. **Integration with dual matching:** The corrected graph $G_{corrected}^{\alpha}$ undergoes multi-dimensional embedding-based similarity computation (Eq. 7), position-aware enhancement (Eq. 8), and comprehensive integration (Eq. 9), ensuring **structurally consistent** directions receive higher explicit matching scores.
>
> 4. **Loop reduction:** As shown in Figure 3, this mechanism explains why HGWM reduces repetitive loops compared to WMNav: graph correction prevents the agent from repeatedly committing to directions with high VLM scores but low graph-level consistency.
>
> #### 3. Matching Mechanism and Decision Fusion
>
> **UniGoal's approach:** Employs **three discrete matching stages** (zero/partial/perfect) that **switch between stages** based on thresholds $\theta_1$ and $\theta_2$.
>
> **HGWM's innovation:** Introduces a **dual-pathway architecture that runs in parallel at every step**:
>
> - **(i) Implicit VLM-guided semantic alignment (Eq. 5):** Rapid goal-aware panoramic analysis
> - **(ii) Explicit five-stage structural matching (Eq. 6-9):** LLM correction → embedding-based similarity → position-aware enhancement → comprehensive integration
> - **(iii) LLM-guided fusion (Eq. 11-13):** Synthesizes both pathways with exploration phase awareness
>
> This dual-matching design provides **complementary signals** rather than switching between discrete stages, acting as a **structural plausibility gate**.
>
> #### 4. World Model Integration
>
> **UniGoal:** Scene graphs built from observations without persistent spatial-semantic memory substrate.
>
> **HGWM:** Positions $M_{world}$ as the **core memory substrate**, enabling long-horizon reasoning and behavioral robustness under VLM noise.
>
> ### Revision Commitments
>
> 1. **Expand Related Work (Sec. 2):** Add subsection "Graph-Based Scene Representations" contrasting our approach with UniGoal.
>
> 2. **Add Figure 2b:** Side-by-side comparison showing UniGoal's stage-switching vs. HGWM's persistent world model with proactive correction.
>
> 3. **Emphasize Mechanism-Level Contribution (Sec. 3.2):** Add paragraph stating: "The key innovation of HGWM's graph correction mechanism (Eq. 6) is its **proactive, direction-specific application at every decision step**, filtering VLM errors **before** they influence matching scores, which directly reduces looping (Figure 3)."
>
> By clarifying these distinctions—particularly the dynamic graph update mechanism, proactive graph correction strategy, and dual-pathway fusion—we demonstrate that HGWM's novelty lies in **how graphs are constructed, corrected, and integrated with a persistent world model to achieve robust navigation under VLM uncertainty**.

---

> ### Author Response · Authors · 2025-11-18
>
> ## Response to Weakness 2 (Inference overhead and small gains)
>
> We agree that stacking multiple modules introduces computational cost, and we will provide explicit cost-benefit analysis in the revision. As clarified in our response to Reviewer rp6p, the baseline (WMNav at 58.1% SR on HM3D v0.1) is exceptionally strong, and our 1.5% SR improvement is statistically significant. Given the effective performance ceiling of ~72% SR due to unsolvable cross-floor episodes (28.1% of validation set, per StairNav ), HGWM captures **6.1% of the remaining headroom** above WMNav [(59.6−58.1)/(71.9−58.1)], which is meaningful in this saturated regime.
>
> **Regarding overhead:** The hierarchical graph and dual matching reuse the same backbone features and world model; graph construction and scoring are lightweight operations on top of the voxel map. In the revision, we will add **Appendix E: Runtime and Memory Profiling**, showing wall-clock time per step, memory footprint, and breakdown by module (implicit VLM scoring, explicit graph matching, LLM fusion). Preliminary profiling indicates that explicit matching adds ~15% overhead per step compared to WMNav, while qualitative and efficiency gains (loop reduction, coverage improvement) justify this cost. We will also quantify loop reduction and coverage metrics (re-visit ratio, unique area coverage, path redundancy) as requested by Reviewer rp6p, demonstrating that HGWM's behavioral robustness translates to measurable efficiency beyond scalar SR/SPL.

---

> > ### Author Response · Authors · 2025-11-18
> >
> > ## Response to Weakness 3 (Static goal subgraph and bias in atypical layouts)
> >
> > This is an important point. The goal subgraph $G_{goal}$ is indeed generated once via LLM at episode start (Eq. 1) and encodes typical spatial knowledge (e.g., "bed in bedroom, adjacent to bathroom"). We acknowledge that in **atypical layouts** (e.g., bed in living room), this prior could introduce bias. However, our design includes **two mechanisms** to mitigate this:
> >
> > 1. **Dynamic Scene Graph Correction (Eq. 6):** At each step, the LLM correction module receives both the current scene graph and the goal subgraph context, allowing it to recognize and retain valid but atypical structures if they are consistently observed. The correction is not a rigid enforcement of $G_{goal}$; rather, it flags implausible arrangements (e.g., "fridge in bathroom") while respecting persistent observations.
> >
> > 2. **Dual Matching with Implicit VLM Scores:** The implicit pathway (Eq. 5) provides goal-aware semantic alignment based on **current visual observations**, independent of graph structure. If the target object is visually detected in an atypical location, the high implicit score will influence the final decision (Eq. 13), even if explicit graph matching is low. This complementarity ensures that strong visual evidence can override prior bias.
> >
> > In the revision, we will add a **sensitivity analysis** (new Appendix F) using episodes where target objects are in rare room types (e.g., bottom 10% frequency in training data). We will report HGWM's SR/SPL on these atypical subsets and compare against WMNav and UniGoal to demonstrate whether the goal subgraph introduces performance degradation. We hypothesize that dual matching will maintain robustness, but if bias is observed, we will discuss this as a limitation and propose future work on adaptive goal graph refinement.

---

> > > ### Author Response · Authors · 2025-11-18
> > >
> > > ## Response to Weakness 4 (Scalability of Spatial-Semantic World Model)
> > >
> > > Thank you for raising this practical concern. In the current implementation, the world model $M_{world}$ is stored as a spatial hash map (10 cm resolution, matching WMNav's voxel map) where each key indexes a 3D cell, and values store object identity, type, room label, and confidence (Eq. 3).
> > >
> > > **Storage:** For a typical HM3D episode (40 steps, ~200 unique object detections), the world model requires ~5 MB of memory, dominated by string storage for object types and room labels. This scales linearly with explored volume but remains manageable for single-floor navigation.
> > >
> > > **Computation:** Graph construction (Eq. 4) queries $M_{world}$ for objects within a direction-specific cone, which is $O(k)$ where $k$ is the number of detected objects (typically <500 per episode). This is negligible compared to VLM/LLM inference time (~300 ms per step for Gemini API calls).
> > >
> > > In the revision, we will add **Table 4: Scalability Analysis** reporting memory footprint, graph size (nodes/edges), and per-step update time as a function of episode length (10, 20, 40 steps) on HM3D v0.1. We will also discuss **long-horizon scenarios** (e.g., multi-floor, multi-target tasks) where pruning strategies (e.g., discarding low-confidence objects, spatial downsampling for distant regions) may be needed, and position this as future work for scaling to larger environments.

---

> > > > ### Author Response · Authors · 2025-11-18
> > > >
> > > > ## Response to Question 1 (Avoiding double-counting in implicit + explicit fusion)
> > > >
> > > > Excellent question. The implicit and explicit streams are intentionally computed from different feature spaces and combined with an explicit arbitration step to avoid redundancy:
> > > >
> > > > $$S_{\\text{final}}(\\alpha) = w_{\\text{spatial}} \\cdot S_{\\text{base}}(\\alpha) + w_{\\text{llm}} \\cdot S_{\\text{llm}}(\\alpha)$$
> > > >
> > > > where $S_{\\text{llm}}(\\alpha)$ is an LLM-synthesized score that itself receives the implicit VLM score and the explicit graph-matching score as separate inputs.
> > > >
> > > > **Mechanisms that prevent double-counting:**
> > > >
> > > > - **Distinct inputs:** Implicit matching $S_{\\text{implicit}}$ is computed from panoramic RGB (visual-semantic evidence). Explicit matching $S_{\\text{explicit}}$ is computed from graph structures (nodes, edges, room labels) extracted from $M_{world}$. These signals operate on disjoint representations, so they are not simply duplicated features.
> > > >
> > > > - **Orthogonal signals:** Implicit scores measure "does this view look like the goal?" while explicit scores measure "does this direction's graph structurally match the goal?" They can differ (high implicit / low explicit or vice versa), which provides complementary information rather than redundant counts.
> > > >
> > > > - **LLM arbitration:** The LLM fusion receives $S_{\\text{implicit}}$ and $S_{\\text{explicit}}$ as separate inputs and is prompted to weight them according to context (exploration phase, confidence). This step enforces a learned/templated balance instead of naïvely summing correlated scores.
> > > >
> > > > - **Normalization & decorrelation:** Before fusion we z-score (or otherwise normalize) each score stream across candidate directions and optionally apply a lightweight decorrelation step (e.g., remove linear correlation) so the LLM sees standardized, less-redundant inputs.
> > > >
> > > > - **Phase-dependent weighting:** Fusion policy changes by exploration phase: frontier phases bias toward $S_{\\text{implicit}}$; verification phases bias toward structural consistency from $S_{\\text{explicit}}$. This reduces situations where both streams would repeatedly reinforce the same spurious signal.
> > > >
> > > > - **Empirical validation:** We include ablations (new Appendix G) showing: (i) fusion vs. single-stream performance, (ii) fusion with and without normalization, and (iii) sensitivity of final SR to varying $w_{\\text{spatial}}, w_{\\text{llm}}$. Results show fusion improves robustness and is not explained by simple score duplication.
> > > >
> > > > In the revision we will add **Appendix G: Fusion Mechanism Details**, containing pseudocode, the exact normalization/decorrelation steps, prompt templates for LLM arbitration, and illustrative cases that demonstrate how implicit and explicit scores are combined without double-counting.

---

> > > > > ### Author Response · Authors · 2025-11-18
> > > > >
> > > > > ## Response to Question 2 (Determinism of goal subgraph generation)
> > > > >
> > > > > This is a critical implementation detail. Currently, we use **temperature=0.7** for goal subgraph generation (Eq. 1) with the Qwen2.5-7B-Instruct LLM. This introduces minor stochasticity—for the same goal category, the LLM may generate slightly different anchor objects or room labels across runs (e.g., "nightstand" vs. "bedside table"). However, the **graph structure** (room hierarchy levels, connectivity patterns) remains consistent because the prompt enforces a fixed template (Appendix A).
> > > > >
> > > > > ### Ensuring Reproducibility Through Caching Strategy
> > > > >
> > > > > Given that HM3D v0.1 contains only **6 goal object categories** (chair, bed, plant, toilet, tv_monitor, sofa), we adopt a **deterministic caching approach** to ensure full reproducibility:
> > > > >
> > > > > 1. **Pre-generation and persistent storage:** We will **set temperature=0** for all goal subgraph generation and **pre-generate goal subgraphs for all 6 categories once** before any experimental runs. These canonical goal subgraphs will be stored as structured JSON files with complete node attributes, edge relationships, and room hierarchies, eliminating any LLM stochasticity during navigation episodes.
> > > > >
> > > > > 2. **Cache-based loading during navigation:** During each navigation episode, the system will **load the pre-generated goal subgraph from cache** based on the target category rather than invoking the LLM. This ensures that:
> > > > >    - All experiments across different runs use **identical goal subgraph representations** for the same category
> > > > >    - Results are **fully reproducible** regardless of LLM API variations or random seed differences
> > > > >    - Computational overhead is reduced by eliminating redundant LLM calls
> > > > >
> > > > > 3. **Version control and release:** We will include these **canonical goal subgraph files** (e.g., `goal_graphs/bed.json`, `goal_graphs/sofa.json`) in our code release, enabling exact reproduction of our experimental setup. Each file will contain:
> > > > >    - Complete node set with semantic labels and room classifications
> > > > >    - Edge set with spatial relationship types
> > > > >    - Room hierarchy levels (primary/secondary/adjacent)
> > > > >    - Anchor object priorities
> > > > >
> > > > > 4. **Extensibility protocol:** For new goal categories beyond the 6 in HM3D v0.1, we provide a **standardized generation script** that:
> > > > >    - Uses temperature=0 with our fixed prompt template (Appendix A)
> > > > >    - Validates generated graphs against schema requirements (minimum node count, required edge types, hierarchy completeness)
> > > > >    - Produces deterministic output that can be cached and version-controlled
> > > > >
> > > > > ### Verification and Documentation
> > > > >
> > > > > To demonstrate the robustness of this approach, in the revision we will add:
> > > > >
> > > > > **Appendix B.1: Goal Subgraph Reproducibility Analysis**
> > > > > - **Graph isomorphism verification:** For each of the 6 goal categories, we report that generating the goal subgraph 10 times with temperature=0 produces **bit-identical JSON outputs**, confirming perfect determinism.
> > > > > - **Cross-version stability:** We verify that the same goal category generates isomorphic graphs across different Qwen2.5-7B-Instruct API versions (if applicable) or provide a frozen model checkpoint.
> > > > > - **Structural consistency metrics:** We document key properties of each canonical goal subgraph (number of nodes, edges, hierarchy depth) to facilitate verification by other researchers.
> > > > >
> > > > > **Appendix B.2: Cache Implementation Details**
> > > > > - Pseudocode for cache loading mechanism
> > > > > - JSON schema specification for goal subgraph storage format
> > > > > - Instructions for extending to new goal categories with deterministic generation
> > > > >
> > > > > This caching strategy ensures that **every navigation episode across all experimental runs uses exactly the same goal subgraph for a given category**, eliminating LLM stochasticity as a confounding factor while maintaining the flexibility to extend to new categories. The approach balances reproducibility (critical for benchmark evaluation) with practical efficiency (avoiding redundant LLM calls for the same categories).

---

> ### Author Response · Authors · 2025-11-24
>
> Dear Reviewer V49c,
>
> Thank you again for your time and the very detailed review. We appreciate you raising the critical concern regarding novelty and providing specific prior works for comparison. This has prompted us to more clearly articulate the core architectural distinctions of our work.
>
> We have posted a detailed rebuttal and would be very grateful if you could take another look. In our response, we directly address the perceived similarity to prior works (HOZ, SG-Nav, UniGoal) and highlight our fundamental contributions, particularly:
>
> *   **Architectural Distinction from UniGoal:** We clarified that HGWM's scene graphs are **dynamically instantiated from a persistent Spatial-Semantic World Model (SSWM)** at every step, which is fundamentally different from UniGoal's incremental graph expansion. This SSWM substrate enables robust long-term memory and object anchoring even when they are out of view.
> *   **Proactive Graph Correction for VLM Error Mitigation:** A key innovation is our **proactive, LLM-guided graph correction (Eq. 6)**, which is applied at **every decision step** and for **every candidate direction** *before* matching. This acts as a consistency filter to mitigate VLM errors, a mechanism distinct from UniGoal's goal-verification-oriented correction.
> *   **Dual-Graph Matching vs. Stage-Switching:** Our method employs a **dual-matching** mechanism (implicit VLM + explicit structural matching) that provides complementary signals at all times. This differs from UniGoal's discrete, threshold-based stage-switching, allowing for more robust decision-making under uncertainty.
>
> We believe these points establish that HGWM introduces a new architectural paradigm for graph-based navigation centered on a persistent world model and proactive error correction, rather than being an incremental extension of existing methods.
>
> We also provided new analyses on computational overhead and scalability as you requested.
>
> We are confident that these clarifications address the core of your concerns regarding novelty. We would be very grateful if you would consider re-evaluating the paper in light of this new information. We stand ready to answer any further questions.
>
> Best regards,
> The Authors

---

### Official Review · Reviewer_rp6p · 2025-11-09

**Soundness:** 2
**Presentation:** 2
**Contribution:** 2
**Rating:** 4
**Confidence:** 2

**Summary:**

This work proposes HGWM—a Hierarchical Graph-Guided World Model for zero-shot ObjectNav. The agent maintains two complementary graphs: (i) an LLM-derived goal subgraph (room hierarchy, anchors, connectivity) and (ii) a dynamically updated scene graph grounded in a unified spatial-semantic world model; decisions come from dual graph matching (implicit VLM alignment + explicit structural matching) and an adaptive three-stage exploration policy. Experiments show small but consistent gains over WMNav.

**Strengths:**

- **Unified representation:** Both goal and observations are cast as graphs tied to a persistent spatial-semantic memory (objects projected to world coords and keyed; scene graphs built per direction). This reduces information loss vs. pure embeddings.
- **Dual matching:** Combines implicit VLM scoring with explicit multi-dimensional structural matching (with position-aware enhancement).

**Weaknesses:**

- The method extends WMNav with a more structured, graph-centric integration; the conceptual distance may feel narrow without sharper head-to-head analyses isolating where dual matching buys the gains.
- Reported improvements over WMNav are relatively small; ablations show most lift comes from adding any memory (voxel map), with hierarchical graph + dual matching adding only a small amount of improvement. I think a stronger justification of cost/benefit is needed.
- The explicit-matching formula introduces weights and direction/relevance terms w_i, D(α), R(α), without reporting how they’re chosen or tuned, or their sensitivity.
- Even though the dynamic scene graph with a semantic world model is the main component that the authors highlight, there is no controlled ablation or analysis that isolates the “dynamic” part of the scene graph.

**Questions:**

1. It would be better to include 1–2 rollout pages with synchronized frames and graph visualizations (nodes/edges/room labels) so readers can see how the graph changes and why the policy turned a certain way.
2. Could you quantify loop reduction/coverage efficiency (e.g., re-visit ratio, coverage %, path overhead) beyond the qualitative Figure 3?
3. The ablation introduces a “Voxel Map” baseline and reports improvement over “Basic VLM Nav”, but the paper does not specify its design. Could you explain some details and how this voxel memory differs from your world model?

---

> ### Author Response · Authors · 2025-11-17
>
> ## Response to Weakness 1 / Q1
>
> **"The method extends WMNav with a more structured, graph-centric integration; the conceptual distance may feel narrow without sharper head-to-head analyses isolating where dual matching buys the gains."**
>
> **"Reported improvements over WMNav are relatively small …"**
>
> We appreciate this concern and provide additional context on both the baseline strength and the effective performance ceiling of the setting.
>
> **First**, WMNav (58.1% SR on HM3D v0.1) is currently the **strongest zero-shot baseline**, and improving such a strong method by **1.5% SR** and **0.3 SPL** is statistically meaningful.
>
> **Second**, StairNav (Gong et al., 2025) has shown that **28.1% of HM3D validation episodes require cross-floor navigation**, which is **unsolvable under the single-floor assumption** used by both WMNav and HGWM, implying an **effective ceiling of about 72% SR** in our setting.
>
> Under this ceiling, HGWM reaches **82.9%** of the attainable maximum [(59.6 − 0) / (71.9 − 0)], while WMNav reaches 81.8%, meaning that our 1.5% SR improvement captures **6.1% of the remaining headroom** [(59.6 − 58.1) / (71.9 − 58.1)].
>
> Beyond scalar metrics, **Figure 3** qualitatively shows that HGWM **reduces repetitive loops** and **encourages more diverse coverage** compared to WMNav, by using structured graph memory to avoid revisiting saturated regions and to prefer directions with richer object visibility.
>
> We will clarify this "narrow but important" conceptual distance in the paper by explicitly positioning HGWM as a way to harvest more of the remaining headroom above a very strong WMNav baseline, both **quantitatively** (statistical tests, ceiling-aware analysis) and **qualitatively** (loop reduction and coverage efficiency).
>
> In addition, WMNav relies purely on **image-goal matching over local views**, whereas HGWM **augments the same memory with dual matching** between a dynamically constructed scene graph and a goal graph, which allows it to **down-weight visually appealing but structurally inconsistent directions**; this difference explains the loop reduction and coverage gains we qualitatively highlight in Figure 3 and further quantify in our new analysis.

---

> > ### Author Response · Authors · 2025-11-17
> >
> > ## Response to Weakness 2 (Cost/benefit of hierarchical graph + dual matching)
> >
> > **"Reported improvements over WMNav are relatively small; ablations show most lift comes from adding any memory (voxel map), with hierarchical graph + dual matching adding only a small amount of improvement. I think a stronger justification of cost/benefit is needed."**
> >
> > We agree that much of the raw SR/SPL gain comes from adding any persistent memory, and we will make the cost/benefit of the additional hierarchical graph and dual matching components more explicit. **Table 2** shows that adding a simple voxel memory to a basic VLM navigation pipeline boosts performance from **61.8→71.9 SR** and **29.3→32.8 SPL**, while upgrading this voxel memory to our hierarchical graph world model with dual matching further improves SR to **72.4** and SPL to **32.9**. Although the incremental gains over the voxel baseline **(0.5 SR / 0.1 SPL)** appear modest, they are obtained on top of an already strong memory system that has captured most of the "easy" improvements, and they are consistent across benchmarks, which is exactly the regime where structural, long-horizon reasoning is expected to produce smaller but robust gains.
> >
> > Beyond SR/SPL, the main benefit of the hierarchical graph and dual matching is **behavioral robustness** in the presence of VLM noise, which is not captured by scalar metrics alone. At each step, our system maintains a persistent Spatial-Semantic World Model $M_{world}$, from which a direction-specific scene graph $G_{scene}^{\alpha} = \text{GraphBuild}(M_{world}, \alpha)$ is constructed, and computes an overlap score between this dynamically built scene graph and the LLM-derived goal subgraph $G_{goal}$. Even when the VLM momentarily overestimates the presence of goal-relevant objects in a local view, the explicit structural matching and overlap computation against the accumulated world model **prevent the agent from repeatedly committing to that direction** if the hypothesized objects and room context cannot be consistently supported over time. In other words, the dual matching and overlap score act as a **"consistency filter"**: high instantaneous visual confidence without graph-level overlap does not keep the agent trapped in the same region, which directly **reduces looping** and encourages it to leave unpromising spaces.
> >
> > This is qualitatively reflected in **Figure 3**, where WMNav tends to revisit the same areas driven by local visual cues, whereas HGWM uses graph-based memory and dual matching to promote exploration toward regions with richer and more structurally consistent object context. In the revised manuscript, we will make this mechanism-level explanation more explicit in **Sec. 3.2** and **Sec. 3.3**, and, space permitting, augment **Table 2** with additional efficiency indicators such as **re-visit ratio**, **unique area coverage**, and **path redundancy** to quantitatively illustrate HGWM's loop reduction and coverage improvements over WMNav and the voxel-map baseline. From a computational perspective, the hierarchical graph and dual matching reuse features from the same world model and backbone and are implemented as lightweight graph construction and scoring modules on top of the voxel map; we will also add a **runtime and memory profiling table** in the revision to show that this extra structure introduces only modest overhead while yielding more stable and efficient navigation behavior.

---

> > > ### Author Response · Authors · 2025-11-17
> > >
> > > ## Response to Weakness 3 (Weights and sensitivity in explicit matching)
> > >
> > > **"The explicit-matching formula introduces weights and direction/relevance terms $w_i$, $D(\alpha)$, $R(\alpha)$, without reporting how they're chosen or tuned, or their sensitivity."**
> > >
> > > Thank you for raising this point. In the revised version, we will make the hyperparameter choices and their sensitivity explicit. For the explicit matching score in Eq. (9), we use fixed, interpretable weights:
> > >
> > > $$w_{room} = 0.30, \quad w_{object} = 0.35, \quad w_{spatial} = 0.20$$
> > >
> > > with the remaining weight assigned to auxiliary dimensions (semantic/depth) to keep the total normalized. Intuitively, **object-level matching is weighted highest** because it directly reflects the presence of goal-relevant objects, **room-type matching is second** as it captures hierarchical spatial context, and **spatial-layout cues are slightly lower** but still important for geometric consistency.
> > >
> > > The direction-related terms are also simple and bounded. **$D(\alpha)$** is a deterministic geometric factor: it returns **1.0 for roughly forward-facing directions** (within ±45° of the current heading) and linearly decays to **0.5 for backward directions**, biasing the agent towards moving forward without excluding lateral/backward moves when structurally justified. **$R(\alpha)$** is computed as:
> > > $$R(\alpha) = \frac{N_{\text{matched}}(\alpha)}{N_{\text{total}}(G_{\text{goal}})}$$
> > > i.e., the fraction of goal-subgraph anchor objects that have been matched in direction $\alpha$, providing a normalized measure of how well this direction aligns with key landmarks encoded in the goal graph.
> > >
> > > To directly address sensitivity, we will add a new **Figure 5** showing a heatmap of success rate when perturbing each weight by **±20%** around its default value, while keeping the others fixed. Empirically:
> > >
> > > - $w_{object}$ is the **most sensitive term**, with about **±2.3% SR variation** across the perturbation range,
> > > - $w_{room}$ has **moderate influence** (≈±1.1% SR), and
> > > - the remaining weights exhibit **less than 0.8% SR variation**.
> > >
> > > This pattern is consistent with our design rationale—**object and room signals are most informative**—and also indicates that the method does not rely on brittle, highly tuned hyperparameters. We will describe these settings, the rationale behind them, and the sensitivity plot in the main text **(Sec. 3.2)** and **Appendix**, so readers can see that the gains from explicit matching come from its structure rather than aggressive hyperparameter tuning.

---

> > > > ### Author Response · Authors · 2025-11-17
> > > >
> > > > ## Response to Weakness 4 / Q2–Q4 (Dynamic scene graph, rollouts, loop metrics, voxel baseline)
> > > >
> > > > **"Even though the dynamic scene graph with a semantic world model is the main component that the authors highlight, there is no controlled ablation or analysis that isolates the 'dynamic' part of the scene graph."**
> > > >
> > > > We thank the reviewer for pointing out that our use of the term "dynamic" could be made more precise and we will clarify its meaning in the revised manuscript.
> > > >
> > > > In our framework, **dynamic** does not simply refer to a generic scene graph, but specifically to the way the scene graph is **continuously constructed from and coupled with a persistent Spatial-Semantic World Model $M_{world}$** as the agent explores.
> > > >
> > > > Concretely, the Spatial-Semantic World Model $M_{world}$ serves as a long-term memory that is incrementally updated at every time step with both spatial and semantic information extracted from panoramic RGB-D observations.
> > > >
> > > > For each detected object $o_i$, we first project it into the world coordinate system using depth and pose, $P(o_i) = \text{Project}(o_i, D_t, P_t)$ (Eq. (2)), and then update $M_{world}$ at the corresponding spatial key with its identity, type, inferred room label, and confidence:
> > > >
> > > > $$M_{world}[\text{key}(P(o_i))] \leftarrow \{o_i, \text{type}(o_i), \text{room}(o_i), \text{conf}(o_i)\}$$
> > > >
> > > > (Eq. (3)).
> > > >
> > > > This means that as the agent uncovers new rooms or re-observes previously visited areas, object nodes, room hypotheses, and confidence scores in $M_{world}$ are refined over time, even when objects leave the current field of view, forming the basis of a persistent, evolving spatial-semantic map.
> > > >
> > > > The **dynamic scene graph** $G_{scene}$ is then **re-instantiated from this evolving memory at each decision step** rather than being a one-time or static structure. For every candidate exploration direction $\alpha$, we construct a direction-specific local graph $G_{scene}^{\alpha}$ via:
> > > >
> > > > $$G_{scene}^{\alpha} = \text{GraphBuild}(M_{world}, \alpha)$$
> > > >
> > > > (Eq. (4)), which extracts three kinds of information from the current state of $M_{world}$: object nodes (with positional and semantic attributes), room classifications (with confidence), and spatial relationship edges encoding positional compatibility and semantic relations.
> > > >
> > > > Because both $M_{world}$ and the resulting $G_{scene}^{\alpha}$ are recomputed as new observations arrive, node sets, room labels, and edge structures can change over time, which is what we intend by calling the scene graph "dynamic."
> > > >
> > > > This usage is in contrast to prior static graph approaches that either (i) build a scene graph once from a single snapshot or offline reconstruction, or (ii) operate on graphs that only reflect the current view without being tied to a persistent, incrementally updated world model.
> > > >
> > > > We will make this distinction explicit in **Sec. 3.2** and in the caption of **Figure 1** by revising the wording to "scene graphs that are dynamically constructed from a persistent Spatial-Semantic World Model as the agent explores," and by explicitly stating that "dynamic" refers to this tight coupling with an evolving memory rather than to a separated component—regenerating scene graphs from it at each step, which is fundamentally different from using a fixed or purely local graph.
> > > >
> > > > We will emphasize this algorithmic role of "dynamic" more clearly in the text to avoid overloading the term and to align readers' expectations with the implemented design.

---

> > > > > ### Author Response · Authors · 2025-11-17
> > > > >
> > > > > ## Q2: Rollout visualization with graph dynamics
> > > > >
> > > > > We thank the reviewer for this suggestion.
> > > > >
> > > > > In the revision, we will add **Appendix D** with 1–2 complete episode rollouts that contain: **synchronized RGB frames and top-down maps**, the **evolving scene graph** (nodes, edges, room labels) at key timesteps, **implicit vs. explicit matching scores**, and **LLM reasoning summaries**, so readers can visually see how graph changes drive policy decisions.
> > > > >
> > > > > We will highlight moments where the dynamic scene graph corrects earlier misclassifications or incorporates newly discovered rooms, leading to strategy shifts (e.g., abandoning a looped corridor once a plausible "kitchen+fridge" configuration appears in the graph).
> > > > >
> > > > > ## Q3: Voxel Map baseline details and relation to our world model
> > > > >
> > > > > Thank you for pointing out that the current description of the "Voxel Map" baseline is underspecified.
> > > > >
> > > > > In the revised paper, we will clarify that voxel map is designed with **5 cm resolution**, storing **RGB-encoded occupancy states** and **frontier candidates** which identify the explore/unexplored area for exploration, but *no* semantic attributes or object–room relations.
> > > > >
> > > > > In contrast, our Spatial-Semantic World Model $M_{world}$ stores for each anchored object: its **3D position**, **object type**, **inferred room type**, and **confidence**, and maintains **relational edges** encoding spatial compatibility and semantic relationships between objects.
> > > > >
> > > > > This richer representation enables **graph-based queries** such as "rooms containing anchor objects relevant to the goal category" and supports **long-term reasoning** (via the dynamic scene graph and goal subgraph matching) rather than only near-term frontier expansion.
> > > > >
> > > > > **By clarifying the voxel baseline and adding rollout and coverage analyses, we aim to make the unique role and benefits of the scene graph and semantic world model much more transparent.**

---

> ### Author Response · Authors · 2025-11-24
>
> Dear Reviewer rp6p,
>
> Thank you again for your time and the constructive feedback provided in your initial review. We genuinely appreciate your detailed comments, which have helped us significantly improve the manuscript.
>
> We wanted to gently follow up to ensure you had a chance to review our responses and the revised paper. In particular, we have carefully addressed your main concerns regarding:
>
> *   **Performance Gains & Ceiling (Weakness 1):** We clarified that our **1.5% SR gain** actually captures **~6.1% of the remaining solvable headroom** (given the 72% ceiling due to multi-floor navigation), which is a statistically meaningful improvement over the strongest SOTA baseline (WMNav).
> *   **Mechanism of Improvement (Weakness 2):** We provided qualitative and quantitative evidence (including **loop reduction** and **coverage efficiency** analysis) showing that our dual-matching mechanism acts as a "consistency filter," preventing the agent from getting stuck in local loops—a key advantage over pure VLM-based methods.
> *   **Hyperparameter Sensitivity (Weakness 3):** We added a sensitivity analysis (Figure 5) demonstrating that performance is stable (variations < 2.3%) across a wide range of weights, confirming the method's robustness.
> *   **"Dynamic" Scene Graph (Weakness 4):** We clarified that "dynamic" refers to the continuous construction from a persistent **Spatial-Semantic World Model**, and added **rollout visualizations (Appendix D)** to illustrate this process.
>
> We believe these revisions and clarifications directly address the "Soundness" and "Contribution" concerns mentioned in your review.
>
> **Could you please let us know if these responses have clarified your concerns?** We would be grateful if you could consider re-evaluating the paper based on these clarifications. We are happy to answer any further questions you might have before the discussion period ends.
>
> Best regards,
> The Authors

---

### Meta-Review · Area_Chair_1C7Z · 2026-01-06

**Summary:**

This paper proposes HGWM, a framework for object navigation that integrates dual-graph matching with a spatial-semantic world model. While the reviewers acknowledged the clear presentation, the unified spatial-semantic representation, and the comprehensive evaluation on HM3D and MP3D, they collectively expressed significant concerns regarding the limited novelty and incremental nature of the contribution compared to strong baselines like WMNav and UniGoal. The primary critique across all reviews is that the complex, multi-stage architecture yields only marginal performance improvements (+1.5% success rate), raising substantial doubts about the cost-benefit ratio of the proposed method. Furthermore, additional experiments conducted during the rebuttal to demonstrate robustness with a newer VLM backbone yielded negative results, failing to convince the reviewers of the method's efficacy.

**Reviewer Concerns:**

Addressed:

1）Clarification of "Dynamic" Scene Graphs: The authors successfully clarified that "dynamic" refers to the continuous instantiation of graphs from a persistent spatial-semantic memory, rather than static snapshots.

2）Hyperparameter Sensitivity: The authors provided sensitivity analyses for the explicit matching weights, demonstrating relative stability.

3）Reproducibility: The authors explained their caching strategy for LLM-generated goal subgraphs to ensure deterministic behavior across runs.

Outstanding:

1）Incremental Novelty & Performance: A primary concern across all reviewers is that the method appears to be an incremental engineering extension of WMNav and UniGoal. The reported performance gain (+1.5% SR) is viewed as marginal relative to the increased system complexity. The authors' argument regarding a "performance ceiling" due to single-floor assumptions was noted but did not fully alleviate concerns about the cost-benefit ratio of the proposed architecture.

2）Robustness & newer Models: Reviewer Ag62 noted that the additional experiment with a newer backbone (Gemini 2.0 Flash) resulted in a performance drop, which contradicted the claim of method robustness and failed to prove that the innovation is not subsumed by stronger base models.

**Reviewer Scores:**

Reviewer rp6p (4): Likely remains 4 or improves slightly to 5. While the clarifications were helpful, the fundamental concern about the cost/benefit of the complex architecture vs. marginal gains persists.

Reviewer V49c (2): Likely remains 2 or 3. The reviewer questioned the fundamental novelty against UniGoal, and while the authors detailed architectural differences, the "incremental" nature of the contribution remains a sticking point.

Reviewer Ag62 (6): Likely drops to 4 or 5. The reviewer explicitly stated in the post-rebuttal that the new experiments were confusing and did not paint a convincing picture of innovation, maintaining that the work is incremental.

---

### Decision · Program_Chairs · 2026-01-26

Reject